

# Method for testing the calibration of acceleration and pressure gauges installed at the ocean bottom

Mikhail Nosov[1,2], Viacheslav Karpov[1,2], Sergey Kolesov[1,2], Kirill Sementsov[1], Hiroyuki Matsumoto[3], and Yoshiyuki Kaneda[4]

[1]Chair of Physics of Sea and Inland Water, Faculty of Physics, M.V.Lomonosov Moscow State University, 119991, Leninskie Gory, Moscow, Russia
[2]Institute of Marine Geology and Geophysics, Far Eastern Branch of Russian Academy of Science, 693022, Nauki 1B, Yuzhno-Sakhalinsk, Russia
[3]Japan Agency for Marine-Earth Science and Technology (JAMSTEC), 237-0061, 2-15 Natsushima, Yokosuka, Japan
[4]Institute of Education, Research and Regional Cooperation for Crisis Management, Kagawa University, 760-8521, 1-1 Saiwai-cho, Takamatsu, Japan

**Correspondence:** Viacheslav Karpov (va.karpov@physics.msu.ru)

**Abstract.** A method is proposed for testing pressure gauges and z-accelerometers, installed in ocean-bottom observatories. The method is based on the linear relationship between variations of the ocean-bottom pressure and the z-acceleration, observed during seismic movements of the bottom within the frequency band of "forced oscillations". Calculation of the boundaries of this frequency band is based on the ocean depth at the observatory site making use of explicit formulae. In the case of correct calibration of the gauges calculation of the ratios of power spectra of bottom pressure variations and the z-accleration within the band of "forced oscillations" yields constant values equal to the square ratio of the total mean pressure and the gravity acceleration. The conditions for application of the proposed method are formulated.

## 1 Introduction

At the beginning of the 21-st century deep-water observatories, equipped with ocean-bottom seismometers (OBS) and pressure gauges (PG), became widespread Suetsugu and Shiobara (2014); Rabinovich and Eblé (2015). Measurements at the ocean bottom are used, in particular, in resolving such important problems as the early warning of earthquakes and tsunamis Titov et al. (2005); Kanazawa (2013); Maeda et al. (2015); Yamamoto et al. (2016); Takahashi et al. (2017). Seismometers installed at the bottom of oceans and seas substantially complement land seismic networks, which contributes significantly to enhancement of the accuracy and speed of the determination of parameters of earthquake sources and, also, opens new possibilities for seismic tomography Webb (1998); Rawlinson et al. (2010); Ranasinghe et al. (2018). Ocean-bottom PGs not only permit to observe tsunami waves, but also other phenomena that manifest themselves in variations of the ocean-bottom pressure (e.g. tide waves, hydroacoustic and seismic waves) Filloux (1983); Levin and Nosov (2016).



The example of two networks of deep-water observatories can be noted: DONET (Dense Oceanfloor Network System for Earthquakes and Tsunamis) in the Nankai Trough area Kaneda (2010) and the S-net (Seafloor Observation Network for Earthquakes and Tsunamis) in the Japan Trench area Kanazawa (2013). Of all the similar networks presently in operation DONET (51 observatories) and the S-net (150 observatories) are the most large-scale systems.

Undoubtedly, the ambitious project SMART (Science Monitoring and Reliable Telecommunication) should be recognized as promising in the field of deep-water measurements; it is aimed at equipping the next generation of trans-oceanic submarine cables with various geophysical sensors, including OBSs and PGs Tilmann et al. (2017); Ranasinghe et al. (2018). Blocks of sensors are to be installed on repeaters located on the submarine cables at intervals of about 50 km. Simple estimation reveals that realization of the project SMART may result in the total number of bottom observatories on the World Ocean floor
increasing up to several thousand.

  Given such a significant number of presently existing and planned deep-water observatories a noticeable amount must be expected of failures of the measuring systems, of errors in the calibration of sensors etc. Moreover, the possibility cannot be excluded of the sensitivity of sensors changing with time or owing to external influences (e.g. the PG being covered with a layer of sediments, a change in the orientation of the OBS axes resulting from a strong nearby earthquake). Access to a measuring
system, situated at a large depth, and/or its replacement always represents a technically complicated and extremely expensive operation. For this reason, it is important to have a way to test the measuring systems, that can be implemented remotely, for example, by an analysis of signals recorded during the next seismic event.

  In recent works Levin and Nosov (2016); Matsumoto et al. (2017); Nosov et al. (2018) variations of the ocean-bottom pressure were shown, in the case of seismic movements of the bottom within a certain frequency band, to be related to the
vertical component of the bottom acceleration by a simple linear dependence. The existence of such a relationship permits to propose a method for testing the calibration of acceleration and pressure gauges located on the ocean floor. The main purpose of the present work consists in description of the essence of this method and of an example of its practical application.

## 2 The relationship between pressure variations and the vertical acceleration

We shall consider variations of the bottom pressure to be the quantity determined by the formula

$$p = P - \overline{P}, \tag{1}$$

where $P$ is the total pressure at the ocean-bottom, $\overline{P}$ is the total pressure averaged over time. The linear relationship between bottom pressure variations and the vertical acceleration component in the case of seismic movements of the ocean-bottom is first mentioned as a theoretical assumption in ref. Filloux (1982). Later, Jean Filloux Filloux (1983) pointed out that such a relationship should not always be observed, but only in the range of low frequencies, when the compressibility of water can
be neglected. Confirmation of Jean Filloux's theoretical assumption on the basis of on-site measurements and correction of the frequency band, in which $p \sim a_z$, turned out to be possible only at the beginning of the 21-st century.

  The first attempt at comparison of signals $p$ and $a_z$, registered at the ocean bottom during a submarine earthquake, was described in ref. Bolshakova et al. (2011). Data obtained by the seafloor observatory Kushiro-Tokachi/JAMSTEC, that registered



the 2003 Tokachi-oki earthquake, permitted to establish a band of frequencies, within which the power spectra of pressure $p$ variations and of the quantity $\rho H a_z$ practically coincided (where $\rho$ is the density of water, $H$ is the ocean depth). Later, the relationship between variations of the bottom pressure and acceleration of the bottom movement was observed repeatedly and studied theoretically by different authors Matsumoto et al. (2012, 2015); Nosov et al. (2015); Levin and Nosov (2016); An et al. (2017); Matsumoto et al. (2017); Nosov et al. (2018). The results of these studies permitted to formulate the theoretical principles forming a basis for the method of testing acceleration and pressure gauges. We shall briefly present these principles.

A compressible layer of water with a free surface in the field of gravity is characterized by a pair of critical frequencies:

$$f_g = 0.366\sqrt{g/H}, \tag{2}$$

$$f_{ac} = c/4H, \tag{3}$$

where $g$ is the gravity acceleration, $c$ is the speed of sound in water. In the case of bottom oscillations with frequencies $f < f_g$ gravity waves arise in the layer of water. Hydroacoustic waves arise in the case of bottom oscillations of frequencies $f > f_{ac}$. In the frequency band $f_g < f < f_{ac}$ no waves arise, while bottom oscillations are only accompanied by forced oscillations of the water layer. In the case of forced oscillations the layer of water moves like a solid-state body inseparably linked with the ocean bottom. In this case, according to Newton's second law, the variations of bottom pressure should be proportional to the vertical component of the bottom acceleration.

$$p = m a_z, \tag{4}$$

where $m$ is the mass of a water column of unit cross section. In the case of a homogeneous ocean mass is expressed via density and depth: $m = \rho H$. In this case we obtain the relationship between pressure and acceleration in the form encountered in most publications Filloux (1982, 1983); Bolshakova et al. (2011); Matsumoto et al. (2012); An et al. (2017)

$$p = \rho H a_z. \tag{5}$$

To determine the quantity $m$ it is not really necessary to make use of data on the density of water and on the ocean depth. It suffices to take advantage of the law of hydrostatics, that relates mass and the mean total pressure at the ocean bottom

$$m = \overline{P}/g. \tag{6}$$

With account of expression (6) formula (4) assumes the following form Nosov et al. (2018):

$$p = \overline{P} a_z/g. \tag{7}$$

The method of testing of the calibration of acceleration and pressure gauges is based on the application of formula (7), which holds true within the frequency band of $f_g < f < f_{ac}$. Formula (7) is extremely convenient for such a test, since, besides measured quantities ($\overline{P}, a_z, p$), it involves a sole physical constant $g$, which is stable in time and, if necessary, can be measured for the installation region of a bottom observatory with a high precision.



## 3   Description of the method and of conditions for its applicability

The frequency band of "forced oscillations" is contained between the critical frequencies $f_g$ and $f_{ac}$, the explicit expressions for which are given by formulae (2) and (3). The dependence of critical frequencies upon the ocean depth is shown in fig. 1. When the depth increases, the "forced oscillations" band becomes narrower and is shifted towards lower frequencies. In the

conditions of our planet this band always exists, even in the case of extremely large ocean depths.

Tests of the calibration of pressure and acceleration gauges can be realized as follows. First, the frequency power spectra of signals $p$ and $a_z$, registered during a submarine earthquake, are calculated. Then, the ratio is determined of these spectra. In the case of correct gauge calibration within the frequency band of "forced oscillations" the ratio of spectra should be a constant value, equal to $(\overline{P}/g)^2$.

The first necessary condition, that provides for the possibility of testing gauges, is the existence in the frequency band of "forced oscillations" of a seismic signal of a level noticeably exceeding the noise level of the measuring equipment. From fig. 1 the lower limit of the band is seen to correspond to quite low frequencies $f_g \sim 0.01$Hz in the case of typical ocean depths (several kilometers). It is known that in order to excite low-frequency seimic waves an earthquake must be sufficiently strong, e.g. see Kasahara (1981).

The second necessary condition for applicability of the method consists in the observatory having to be situated on a flat horizontal segment of the ocean bottom far from steep underwater slopes, formula (7) holds valid in such cases. In the case of insignificant slopes of the bottom in the region where the observatory is installed a more general formula is valid Nosov et al. (2018):

$$p = \frac{\overline{P}}{g}\left(\frac{\partial H}{\partial x}a_x + \frac{\partial H}{\partial y}a_y + a_z\right), \tag{8}$$

where $a_x$ and $a_y$ are horizontal components of the acceleration. From formula (8) it is possible to conclude that the level of the contribution of horizontal movements of the bottom to pressure variations is determined by the scalar product of the depth gradient and the vector of horizontal accelerations. Moreover, if an observatory is situated on a nearly horizontal bottom, a certain contribution to pressure variations may be due to horizontal seismic movements of nearby steep underwater slopes. But this contribution rapidly decreases exponentially with the distance. Therefore, the influence of underwater slopes located at

distances exceeding 1-2 ocean depths from the ocean-bottom observatory can be neglected.

The third necessary condition for applicability of the method is actually technical. The whole row of data $T_{max}$ must provide for absolute resolution of the frequency $f_g$. From basic principles of spectral analysis it is known that the condition $T_{max} > 1/f_g$ must be fulfilled. But, in practice, obtaining consistent estimates of power spectra, for instance, applying Welch averaging requires the fulfilment of a more rigid condition: $T_{max} \gg 1/f_g$.



**Figure 1.** Critical frequencies $f_g$ and $f_{ac}$, limiting the band of "forced oscillations", as functions of the ocean depth





## 4 An example of application of the method

We shall demonstrate performance of the method taking advantage of the example of the Great East Japan Earthquake (Mw = 9.0, 2011-03-11, 05:46 UTC), which was recorded by ten DONET observatories, that were in operation in March, 2011. The mutual disposition of the earthquake epicenter and the DONET system is shown in fig. 2.

Each of the DONET observatories is equipped with a PG and an OBS situated at a distance inferior to 10 m from each other. The frequency of digitizing data from the PGs was 10 Hz. The initial sampling frequency of seismic data was 200 Hz. The accelerograms were resampled to the frequency 10 Hz. In further analysis, we shall use the 30-min records (from the beginning of the earthquake), obtained by the PGs and the OBSs (z-accelerometers). Examples of signals, registered by DONET observatories during the Great East Japan Earthquake, are presented in ref. Nosov et al. (2015); Levin and Nosov

(2016); Matsumoto et al. (2017); Nosov et al. (2018).

At the *first* stage PG records were used for calculating the quantity $\overline{P}$, that was determined as the simple arithmetic average of values of the total pressure at the ocean bottom, $P$, during the time interval $T_{max} = 30\ min$. At the *second* stage formula (1) was used in determining variations of the bottom pressure $p$, and their power spectrum was calculated. The power spectrum was calculated applying Welch's averaging method Bendat and Piersol (2010). The time series was divided with the aid of the

Hann window into 7 segments with a 50% overlap. The size of a segment amounted to 4096 readouts. The size (length) of the segment of 409.6 s makes possible guaranteed resolution of the frequency $f_g$, which for a typical installation depth of the DONET observatories ($H \approx 2000\ m$) amounted to $f_g \approx 0.025\ Hz$. At the *third* stage calculation was performed of the power spectrum of the vertical component of acceleration, $a_z$. The calculation technique was absolutely equivalent to the one applied in calculating the power spectrum of pressure variations. At the conclusive *fourth* stage the ratio of spectra was determined.

The confidence interval for the ratio of spectra was calculated by the technique described in ref. Shin and Hammond (2008).

The ratios between spectra are presented in fig. 3 by blue curves. The horizontal dashed line shows the level corresponding to the value $(\overline{P}/g)^2$. The vertical dashed lines show the frequencies $f_g$ and $f_{ac}$, that determine the band of "forced oscillations". For calculating the values of $f_g$ and $f_{ac}$ use was made of the ocean depths at the installation sites of the DONET observatories (the values are given in the figure), the speed of sound in water $c = 1500\ m/s$ and the gravity acceleration $g = 9.8\ m/s^2$.

From fig. 3 it is seen that for observatories A02, A03, A04, B05, B08, D16, E17 the blue curve in the low-frequency band of "forced oscillations" corresponds with quite a good precision to the level $(\overline{P}/g)^2$. This fact is evidence for correct calibration of gauges of the observatories indicated. In the case of observatories B06 and C09, situated in the region of steep underwater slopes, significant deviations of the blue curve from the level, prescribed by theory, are noticeable within the whole band of forced oscillations. Calibration tests cannot be performed for such stations.

At the high-frequency end of the forced oscillation band deviations of the blue curve from the level $(\overline{P}/g)^2$ are observed in all cases. The reason for deviations is explained by manifestations of the compressibility of water. A detailed explanation of this effect is presented in ref. Nosov et al. (2018).

The situation with observatory E18 deserves a special comment. This observatory is situated on a horizontal segment of the bottom far from underwater slopes. But, here, the blue curve is essentially higher than the level $(\overline{P}/g)^2$. This points to an



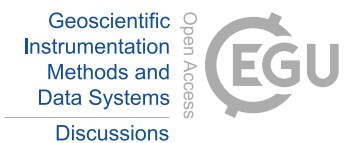

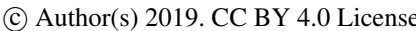

**Figure 2.** Mutual disposition of DONET stations (black triangles) and epicenter of the Great East Japan Earthquake (asterisk), that occurred on 2011-03-11, 05:46 UTC. The insert shows the bottom relief in the region, where the DONET observatories are installed. The isobaths are drawn in steps of 100 m in accordance with JODC-Expert Grid data for Geography (J-EGG500). The radii of the circular marks are equal to the ocean depths at the observatory sites





**Figure 3.** Ratio of power spectra of pressure variations and of the vertical component of acceleration, registered by the PGs and OBSs of ten DONET observatories during the Great East Japan Earthquake. The horizontal dashed line shows the level $(\overline{P}/g)^2$. The vertical lines show the location of critical frequencies $f_g$, $f_{ac}$



error in calibration of the pressure gauge or of the z-accelerometer. In the case considered it was possible to compare signals registered by observatory E18 and by other nearby DONET observatories Nosov et al. (2018). The results of such a comparison permit to conclude that erroneous calibration was performed of precisely the z-accelerometer of E18.

## 5 Conclusions

A method is developed for testing the calibration of acceleration and pressure gauges installed at the ocean bottom. Tests are performed by comparison of the power spectra of signals registered by the gauges during an earthquake. The method is based on the linear relationship between pressure variations and the vertical component of acceleration of the bottom movement, observed within the frequency band of "forced oscillations", and is a direct consequence of Newton's 2-nd law. The boundaries of the "forced oscillations" band are calculated from the ocean depth applying explicit formulae. Besides rows of data, obtained

by the ocean-bottom measuring devices, application of the method requires knowledge of a sole physical constant - the gravity acceleration.

    The *first* condition that makes application of the method possible imposes restrictions on the choice of seismic events. The earthquake must be sufficiently strong for the level of signals, registered by the PG and the z-accelerometer within the frequency band of "forced oscillations", to be noticeably superior to the level of noise. The second necessary condition for application of

the method consists in installation of the bottom registering devices on a horizontal segment of the ocean bottom at a distance exceeding 1-2 ocean depths from steep submarine slopes.

*Data availability.* The DONET data are obtained according to the Implementing Agreement between Faculty of Physics of Lomonosov Moscow State University and JAMSTEC, which does not imply transfer to a third Party.

*Author contributions.* Mikhail Nosov played a leading role in this study, suggested the main idea, provided physical interpretation of data

processing results; Viacheslav Karpov carried out spectral analysis, prepared all the illustrations and typesetted the manuscript; Sergey Kolesov preprocessed the data, downsampled accelerograms; Kirill Sementsov processed three-component acceleration signals and estimated contribution of horizontal acceleration components; Hiroyuki Matsumoto extracted DONET data, provided information on how to transform signals to physical units; Yoshiyuki Kaneda provided data transfer and communication between Russian and Japanese teams.

*Competing interests.* The authors declare that they have no competing interests.

*Acknowledgements.* We are grateful to JAMSTEC and JODC for the data that were kindly placed at our disposal.





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
