# Peer review of "Method for testing the calibration of acceleration and pressure gauges installed at the ocean bottom"

_Geoscientific Instrumentation, Methods and Data Systems, 2019_

## Referee Comment (RC1) · Anonymous Referee #1 · 8 Aug 2019

[General comments] This article proposes a way for testing accelerometers and pressure sensors of seafloor cabled observatories by taking ratio between power spectra of acceleration and pressure records. The idea is very simple but its background theory is well established. It looks that the proposed method can identify errors in records either of accelerometers or pressure gauges effectively. However, this idea was already pointed out by early work of the authors group (Nosov et al., EPS, 2018). I could not find any significant advancement since the previous paper, in the present manuscript. The dataset (accelerograms and pressure records obtained by DONET in Japan) for testing the proposed method is completely identical to the previous work, in which the relation between the seafloor acceleration and pressure is proved by analyzing the field
observations. The results presented here is not new at all, even the previous paper did not show the power spectra ratios explicitly. Therefore, I have very negative feeling about the novelty of the presently submitted manuscript and would not recommend the editor to accept it for publication.

I would suggest to the authors that they develop discussion on applicability of the method in practice. As the authors concluded, the method works only when the system records seafloor motions strong enough. Quantitative elaboration of this aspect may increase the value of this work. How large ground shaking would be necessary for the test with this method? How often can we expect such strong shaking is observed by a certain seafloor cable system? The limitation does not come only from the amplitudes but from the durations of the signals, as the authors also pointed out. What would be the minimum length of records to assess if the sensors work properly? Since the duration is related to the sizes (magnitudes) of earthquakes, this estimation is also important to know how often we can make the test using the method proposed here. In other words, it is very hard to see if the proposed method is practically useful to diagnose seafloor sensors remotely without the discussion of this kind. The systems deployed along subduction zones may have good chances but I'm not sure if the method will be useful for SMART project, introduced in the manuscript, which may be placed on low-seismicity areas.

[Specific comments] A textbook written by Saito [2019]\* would be cited in the explanation of theoretical background (section 2). The textbook gives comprehensive description regarding the ocean acoustic waves and the relation between bottom pressure and acceleration. \* Satio, T., Tsunami generation and propagation, Springer, doi:10.1007/978-4-431-56850-6, 2019.

The meaning of a constant 0.366 appearing in equation (3) should be explained briefly. Although the physical meaning of the equation is explained, it is not clear why the coefficient must be 0.366 in this definition.
I'm curious about the treatment of tide variations in pressure data. Tide variations are well outside of the frequency band for taking spectral ratios, but it must affect the averaged total pressure (P-bar), if not removed from the records.
Interactive

comment

---

## Referee Comment (RC2) · Anonymous Referee #2 · 8 Aug 2019

The manuscript "Method for testing the calibration of acceleration and pressure gauges installed at the ocean bottom" by Nosov et al. presents a method to remotely assess qualities of the pressure gauges and the vertical accelerometers installed at the deep-sea observatories. The manuscript first describes the theoretical basis and the calibration procedure. Then the authors demonstrate its performance from an application to the records of the 2011 Japan Earthquake.

This work deals with an important problem of on-site remote assessments on the deep-sea observatories. I can understand the theoretical basis. The calibration procedure is sufficiently summarized to allow the reproduction. The application is valid. I can read

the manuscript smoothly.

However, some of the explanation on the theoretical basis seem inadequate. I think the authors should perform more extensive analyses and discussions from some important viewpoints, since I could not find the significant expansion of the information already available on the same author's previous paper. Overall, I cannot recommend the current manuscript for publication, but it could be potentially be acceptable after a major revision.

Major comments:

[P.6 LL. 25–26] I suggest the authors discuss more in detail why/how the authors concluded the result had "quite a good precision". If the vertical bars in Figure 3 are the confidence interval (although no explanation is shown), the uncertainties seem too large to immediately conclude as the authors did.

[P. 6 LL. 27–29] I suggest that the authors discuss how to get over the difficulty in the calibration of the deepest observatories located at the trench slope region. Probably the authors can check whether an incorporation of the horizontal acceleration effects improves the calibration or not.

[P.9] I strongly suggest the authors conduct some extensive analyses from some important viewpoints. The authors can perform the application of the method to other major earthquakes to discuss temporal changes of the sensor performance. It may also be valuable to confirm the lowest magnitudes to which the calibration can be applied. This is important to discuss the feasibility of the real-time monitoring since the major events much less frequently occur. In addition, the authors can discuss advantages and disadvantages of the proposed method compared to other approaches, such as the comparison between the different types of seismometers installed at the same site.

Minor comments:

[P.2 LL.11–17] There must be many papers dealing with the calibration methods for onshore seismometers whereas there are much less researches for offshore deep-sea instruments. I suggest the authors refer to the previously proposed approaches in the introduction.

[P.3 L.8] I suggest the authors explain how "0.366" was derived.

[P.4 L.7] The authors can remove "submarine". Large inland earthquakes must generate significant seismic waves in the seafloor observatory.

[P.4 LL. 16–20, 22–23] These sentences confused me. If the slope is insignificant or nearly horizontal, the horizontal effects will be small to be ignored. But it seems that what the authors are saying is different. Please explain carefully.

[P. 4 LL. 23–25] Please explain how "rapid decreases exponentially" and "1-2 ocean depths" are derived and/or cite the appropriate reference.

[P.8 L. 33 – P.9 L. 3] This paragraph is unkind to the readers. To completely understand what the authors are saying, I had to refer back to Nosov et al. (2018). I suggest the authors briefly explain the results of the analysis performed in Nosov et al. (2018) and describe how the authors conclude the vertical accelerometer is in worse condition.

[P.9 LL. 1–2] It is likely that Nosov et al. (2018) compares the E18 spectrum only to E17. Is it OK to use plural form here?

---

## Author Comment (AC1) · 24 Sep 2019

We express our sincere gratitude to Anonymous Referee #1 for attentive reading of the manuscript, understanding of manuscript's essence and valuable comments. Our responses to all the comments are listed below.

Anonymous Referee #1

[General comments] This article proposes a way for testing accelerometers and pressure sensors of seafloor cabled observatories by taking ratio between power spectra of

acceleration and pressure records. The idea is very simple but its background theory is well established. It looks that the proposed method can identify errors in records either of accelerometers or pressure gauges effectively. However, this idea was already pointed out by early work of the authors group (Nosov et al., EPS, 2018). I could not find any significant advancement since the previous paper, in the present manuscript. The dataset (accelerograms and pressure records obtained by DONET in Japan) for testing the proposed method is completely identical to the previous work, in which the relation between the seafloor acceleration and pressure is proved by analyzing the field observations. The results presented here is not new at all, even the previous paper did not show the power spectra ratios explicitly. Therefore, I have very negative feeling about the novelty of the presently submitted manuscript and would not recommend the editor to accept it for publication.

The Referee is only partially right here. In our paper (Nosov et al., EPS, 2018; page 6, right column, the last paragraph) it is indeed mentioned that the revealed relationship between pressure and acceleration can be used for testing the calibration precision of sensors in situ. However the methods is not introduced in the EPS paper. Obviously, it is next to impossible to squeeze the description of the method into one paragraph! The method for testing accelerometers and pressure sensors - including description of all the necessary stages - is developed in present our manuscript. So, we are strongly disagree with Referee's conclusion regarding the absence of novelty.

I would suggest to the authors that they develop discussion on applicability of the method in practice. As the authors concluded, the method works only when the system records seafloor motions strong enough. Quantitative elaboration of this aspect may increase the value of this work. How large ground shaking would be necessary for the test with this method? How often can we expect such strong shaking is observed by a certain seafloor cable system? The limitation does not come only from the amplitudes but from the durations of the signals, as the authors also pointed out. What

would be the minimum length of records to assess if the sensors work properly? Since the duration is related to the sizes (magnitudes) of earthquakes, this estimation is also important to know how often we can make the test using the method proposed here. In other words, it is very hard to see if the proposed method is practically useful to diagnose seafloor sensors remotely without the discussion of this kind.

We thank Referee for these fruitful ideas. The related discussion will be developed in the revised manuscript.

The systems deployed along subduction zones may have good chances but I'm not sure if the method will be useful for SMART project, introduced in the manuscript, which may be placed on low-seismicity areas.

There are no problems with the so-called "low-seismicity areas". We have to recall here that seismic events with magnitudes greater than 5 are strong enough to be detected by a seismograph and/or pressure gauge anywhere in the world except earthquake shadow zones (http://www.isc.ac.uk/).

[Specific comments] A textbook written by Saito [2019]* would be cited in the explanation of theoretical background (section 2). The textbook gives comprehensive description regarding the ocean acoustic waves and the relation between bottom pressure and acceleration. * Satio, T., Tsunami generation and propagation, Springer, doi:10.1007/978-4-431-56850-6, 2019.

This idea is reasonable. We shall refer the textbook in the revised version of manuscript.

The meaning of a constant 0.366 appearing in equation (3) should be explained briefly. Although the physical meaning of the equation is explained, it is not clear why the

coefficient must be 0.366 in this definition.

Brief explanation here involves just a couple of formulas:

1. $1/\cosh(k_0) = 0.01$ – from this equation we get the root $k_0$;

2. $\sqrt{k_0 \tanh(k_0)}/2\pi = 0.366$ – this formula is from the dispersion relation for gravity waves.

We are sure that such an explanation looks much more nebulous than simply the value of 0.366 together with necessary reference. Unfortunately, the meaning of the constant 0.366 can not be explained briefly. It requires at least the following points: (1) description of the problem of tsunami generation by dynamical displacement of ocean-bottom, (2) analytical solution to this problem – rather cumbersome formulas, (3) interpretation of the solution (preferably illustrated by a figure). We have already provided this explanation repeatedly, for example in the monograph "Physics of Tsunamis" by Levin&Nosov 2016 (P. 211-213). It is senseless to repeat over and over again descriptions that had been already published.

I'm curious about the treatment of tide variations in pressure data. Tide variations are well outside of the frequency band for taking spectral ratios, but it must affect the averaged total pressure (P-bar), if not removed from the records.

We do not think it is necessarily to detide the records. At first sight, the influence of the tide should really negligible ~0.1% because at large ocean depth (~1000 m) tide amplitude is about 1 m. In the revised version of manuscript we shall provide a more detailed comment regarding this problem.

---

## Author Comment (AC2) · 24 Sep 2019

We express our sincere gratitude to Anonymous Referee #2 for attentive reading of the manuscript, understanding of manuscript's essence and valuable comments. Our responses to all the comments are listed below.

Anonymous Referee #2

 The manuscript "Method for testing the calibra-

tion of acceleration and pressure gauges installed at the ocean bottom" by Nosov et al. presents a method to remotely assess qualities of the pressure gauges and the vertical accelerometers installed at the deep-sea observatories. The manuscript first describes the theoretical basis and the calibration procedure. Then the authors demonstrate its performance from an application to the records of the 2011 Japan Earthquake.

This work deals with an important problem of on-site remote assessments on the deep-sea observatories. I can understand the theoretical basis. The calibration procedure is sufficiently summarized to allow the reproduction. The application is valid. I can read the manuscript smoothly.

However, some of the explanation on the theoretical basis seem inadequate. I think the authors should perform more extensive analyses and discussions from some important viewpoints, since I could not find the significant expansion of the information already available on the same author's previous paper. Overall, I cannot recommend the current manuscript for publication, but it could be potentially be acceptable after a major revision.

Major comments:

[P.6 LL. 25–26] I suggest the authors discuss more in detail why/how the authors concluded the result had "quite a good precision". If the vertical bars in Figure 3 are the confidence interval (although no explanation is shown), the uncertainties seem too large to immediately conclude as the authors did.

Yes, this fragment needs a revision. "a good precision" will be replaced by some quantitative measure. The vertical bars in Figure 3 are the 95% confidence intervals. In the revised version of manuscript we shall describe how the intervals are calculated.

[P. 6 LL. 27–29] I suggest that the authors discuss how to get over the difficulty in the calibration of the deepest observatories located at the trench slope region. Probably the authors can check whether an incorporation of the horizontal acceleration effects improves the calibration or not.

The suggestion is very reasonable. We shall discuss this issue in revised manuscript. In our opinion the difficulty can not be overcome because pressure variation p are related to acceleration components $(a_x, a_y, a_z)$ as follows: $p = C_x a_x + C_y a_y + a_z$, where $C_x$ and $C_y$ are constants. In the case of flat horizontal bottom $C_x = 0$, $C_y = 0$, so a conclusion on the correctness of calibration of acceleration and pressure gauges can be made. In the case of sloping bottom $C_x$ and $C_y$ take nonzero values. Thus, it is not possible to reveal which accelerometer ($x$, $y$ or $z$) is corrupted.

[P.9] I strongly suggest the authors conduct some extensive analyses from some important viewpoints. The authors can perform the application of the method to other major earthquakes to discuss temporal changes of the sensor performance. It may also be valuable to confirm the lowest magnitudes to which the calibration can be applied. This is important to discuss the feasibility of the real-time monitoring since the major events much less frequently occur. In addition, the authors can discuss advantages and disadvantages of the proposed method compared to other approaches, such as the comparison between the different types of seismometers installed at the same site.

In present study we use only limited (1-week) DONET data-set. This is why we can deal with very limited number of seismic events. The DONET data-set was obtained according to the Implementing Agreement between Faculty of Physics of Lomonosov Moscow State University and JAMSTEC. The Referee's idea regarding study of temporal changes of the sensor performance is certainly very interesting but it can not be realized in near future (new Agreement etc.).

Here it is worthwhile to recall the main goal of our present manuscript. Recently we discovered the relationship between pressure and acceleration that exists under certain conditions (Nosov et al., EPS, 2018). Understanding physics, on the base of this relationship, we suggest new method for testing the calibration. Comparative analysis of newly suggested method with other approaches or study of temporal changes of the sensor performance is beyond of our intensions. These studies deserves a couple separate papers at least.

As for the lowest magnitudes to which the calibration can be applied, in revised manuscript we shall discuss this topic, provide some quantitative estimations (on the base of dependence "corner-frequency vs Mw") and demonstrate a couple of examples (aftershocks of the 2011 Tohoku earthquake).

Minor comments:

[P.2 LL.11–17] There must be many papers dealing with the calibration methods for onshore seismometers whereas there are much less researches for offshore deep-sea instruments. I suggest the authors refer to the previously proposed approaches in the introduction.

We shall do our best to refer such papers. We shall appreciate it if Referee kindly let us know the most important papers.

[P.3 L.8] I suggest the authors explain how "0.366" was derived.

Unfortunately, the meaning of the constant 0.366 can not be explained briefly. It requires at least the following points: (1) description of the problem of tsunami generation by dynamical displacement of ocean-bottom, (2) analytical solution to this problem – rather cumbersome formulas, (3) interpretation of the solution (preferably illustrated by a figure). We have already provided this explanation repeatedly, for example in the

monograph "Physics of Tsunamis" by Levin&Nosov 2016 (P. 211-213). It is senseless to repeat over and over again descriptions that had been already published. Moreover, present manuscript is devoted to the method, but not to its theoretical background.

[P.4 L.7] The authors can remove "submarine". Large inland earthquakes must generate significant seismic waves in the seafloor observatory.

Exactly so! We shall remove "submarine".

[P.4 LL. 16–20, 22–23] These sentences confused me. If the slope is insignificant or nearly horizontal, the horizontal effects will be small to be ignored. But it seems that what the authors are saying is different. Please explain carefully.

The text will be revised.

[P. 4 LL. 23–25] Please explain how "rapid decreases exponentially" and "1-2 ocean depths" are derived and/or cite the appropriate reference.

These points will be explained.

[P.8 L. 33 – P.9 L. 3] This paragraph is unkind to the readers. To completely understand what the authors are saying, I had to refer back to Nosov et al. (2018). I suggest the authors briefly explain the results of the analysis performed in Nosov et al. (2018) and describe how the authors conclude the vertical accelerometer is in worse condition.

We shall do our best to provide more detailed and clear explanation.

[P.9 LL. 1–2] It is likely that Nosov et al. (2018) compares the E18 spectrum only to E17. Is it OK to use plural form here?

We are agree. Of course not plural.